# MYH3-associated distal arthrogryposis zebrafish model is normalized with para-aminoblebbistatin

Julia Whittle[1],[†] , Lilian Antunes[1],[†], Mya Harris[2], Zachary Upshaw[2], Diane S Sepich[3], Aaron N Johnson[3], Mayssa Mokalled[3], Lilianna Solnica-Krezel[3], Matthew B Dobbs[4] & Christina A Gurnett[1,2,5,*]

## Abstract

Distal arthrogryposis (DA) is group of syndromes characterized by congenital joint contractures. Treatment development is hindered by the lack of vertebrate models. Here, we describe a zebrafish model in which a common *MYH3* missense mutation (R672H) was introduced into the orthologous zebrafish gene *smyhc1* (*slow myosin heavy chain 1*) (R673H). We simultaneously created a *smyhc1* null allele (*smyhc1⁻*), which allowed us to compare the effects of both mutant alleles on muscle and bone development, and model the closely related disorder, spondylocarpotarsal synostosis syndrome. Heterozygous *smyhc1*^R673H/+ embryos developed notochord kinks that progressed to scoliosis with vertebral fusions; motor deficits accompanied the disorganized and shortened slow-twitch skeletal muscle myofibers. Increased dosage of the mutant allele in both homozygous *smyhc1*^R673H/R673H and transheterozygous *smyhc1*^R673H/− embryos exacerbated the notochord and muscle abnormalities, causing early lethality. Treatment of *smyhc1*^R673H/R673H embryos with the myosin ATPase inhibitor, para-aminoblebbistatin, which decreases actin–myosin affinity, normalized the notochord phenotype. Our zebrafish model of *MYH3*-associated DA2A provides insight into pathogenic mechanisms and suggests a beneficial therapeutic role for myosin inhibitors in treating disabling contractures.

**Keywords** contracture; hypercontractile; muscle; myosin; notochord
**Subject Categories** Genetics, Gene Therapy & Genetic Disease; Musculoskeletal System

## Introduction

Distal arthrogryposis (DA) describes a group of congenital musculoskeletal syndromes characterized by contractures in the joints of the hands and feet. Classification systems currently describe ten closely related DA subtypes, the most severe of which is Freeman-Sheldon syndrome (also called distal arthrogryposis, type 2A [DA2A]). Children born with DA2A present with characteristic contractures of the hands, clubfeet, and facial contractures. They also often develop scoliosis (Toydemir *et al*, 2006; Beck *et al*, 2013). Complications of DA, particularly DA2A, include difficulty eating, respiratory complications, and impaired speech and mobility. Currently, supportive treatments including bracing, physical therapy, and surgical intervention are the only therapeutic options for DA patients; however, these therapies are often suboptimal (Stevenson *et al*, 2006; Boehm *et al*, 2008).

The most common genetic causes of DA are autosomal dominant missense mutations in the *MYH3* gene, encoding the embryonic myosin heavy chain (MyHC) that is expressed first during slow skeletal muscle development. *MYH3* expression peaks during fetal development, and is significantly downregulated after birth (Chong *et al*, 2015; Schiaffino *et al*, 2015; Cameron-Christie *et al*, 2018). *MYH3* mutations have been identified in multiple DA syndromes, including distal arthrogryposis, type 1 (Alvarado *et al*, 2011), DA2A, and distal arthrogryposis type 2B (DA2B) (also called Sheldon-Hall syndrome) (Toydemir *et al*, 2006; Beck *et al*, 2013). *MYH3* mutations have additionally been identified in patients with multiple pterygium syndrome (Chong *et al*, 2015), and spondylocarpotarsal synostosis syndrome, which is associated with vertebral, carpal, and tarsal fusions in addition to joint contractures (Carapito *et al*, 2016; Zieba *et al*, 2017; Cameron-Christie *et al*, 2018).

MyHCs are large, dimerizing, ATP-dependent motor proteins that bundle together in thick filaments to drive muscle contraction. The most common DA-causing *MYH3* mutations cluster in the ATPase region of the MyHC motor domain (Toydemir *et al*, 2006; Fitts, 2008). The dominant inheritance pattern of DA mutations implies a hypermorphic, neomorphic, or antimorphic nature. The functional effects of DA-associated *MYH3* missense mutations include slowing the muscle relaxation time and prolonging the muscle fiber contracted state (Racca *et al*, 2015; Walklate *et al*, 2016). While the vast majority of described DA mutations are dominant missense variants, the recent identification of autosomal recessive *MYH3* variants in spondylocarpotarsal synostosis syndrome suggests that some mutations may

1   Department of Neurology, Washington University in St. Louis, St. Louis, MO, USA
2   Department of Orthopedic Surgery, Washington University in St. Louis, St. Louis, MO, USA
3   Department of Developmental Biology, Washington University in St. Louis, St. Louis, MO, USA
4   Paley Institute, West Palm Beach, FL, USA
5   Department of Pediatrics, Washington University in St. Louis, St. Louis, MO, USA
    *Corresponding author. Tel: +1 (314) 454 6120; E-mail: gurnettc@wustl.edu
    †These authors contributed equally to this work

also contribute to disease pathogenesis through a loss of function or hypomorphic mechanism (Cameron-Christie et al, 2018).

Understanding the mechanism by which dominant or recessive *MYH3* mutations cause contractures or bony fusions has been limited by poor access to human tissue, particularly during early development when the gene is most highly expressed. Analysis of muscle biopsies from adults with the *MYH3* R672C mutation, which is one of the most common recurrent variants causing DA2A and DA2B (Toydemir et al, 2006), revealed increased relaxation time and impaired sarcomeric cycling (Racca et al, 2015). Recombinantly expressed *MYH3* R672C, R672H, and T178I mutations in cultured cells also caused marked abnormalities in molecular kinetic properties including slower cycling time (Walklate et al, 2016). While the molecular effects of pathogenic *MYH3* mutations have been modeled in *Drosophila* (Rao et al, 2019), mutations have yet to be studied in vertebrate models where their impact on muscle and skeletal development can be concurrently assessed.

To address the need for vertebrate models of DA, we created a zebrafish line in which the common human DA2A *MYH3* mutation, R672H, was precisely edited into the corresponding amino acid of the *slow myosin heavy chain 1* gene (smyhc1$^{R673H}$ [stl583]) (Roy et al, 2001). This gene is expressed in the somites as early as 5–9 somite stages, and in the lateral slow muscle of the trunk, as evidenced by previously conducted *in situ* hybridization (Rauch et al, 2003; Bessarab et al, 2008; Li et al, 2020). We simultaneously created a smyhc1 null allele (smyhc1$^{-}$ [stl582]), which allowed us to compare the effects of both mutant alleles on slow skeletal muscle and bone development.

Remarkably, the smyhc1$^{-}$ allele in the homozygous state caused transient muscle abnormalities in larvae and late-onset scoliosis, whereas the smyhc1$^{R673H}$ allele resulted in notochord kinks, abnormally shortened muscle, and vertebral fusions with the severity of the phenotype correlating with allele dosage. The early notochord phenotype was suppressed or normalized with para-aminoblebbistatin, a drug that reduces myosin–actin affinity by inhibiting myosin ATPase activity, which directly inhibits muscle contraction (Várkuti et al, 2016). Because removing muscle contraction suppressed the abnormal notochord phenotype, we interpret this result as supporting our hypothesis that the smyhc1$^{R673H}$ allele causes hypermorphic muscle contraction that creates excessive tension on the developing zebrafish notochord. This secondarily results in kinking of the notochord that contributes to later vertebral fusions. Furthermore, by showing that the smyhc1$^{R673H}$ zebrafish model replicates key aspects of the congenital disorder that can be normalized with myosin inhibitors, we have demonstrated its value for evaluating therapeutics for use in human DA patients.

# Results

## smyhc1$^{R673H}$ is an autosomal dominant mutation that acts in a dose-dependent manner

To study the mechanism by which *MYH3* mutations cause DA, we genetically engineered a mutant zebrafish line in which a single nucleotide substitution was introduced via homologous recombination into exon 16 using a donor oligonucleotide and TALENs (Fig 1A–C). The resultant zebrafish smyhc1$^{R673H}$ missense substitution is orthologous to a common *MYH3* mutation causing DA2A in

humans, R672H (Fig 1A and B). We concurrently generated a zebrafish smyhc1$^{-/-}$ line containing a frame-shifting seven base pair deletion that is predicted to result in a premature truncation in exon 16 and transcript loss by nonsense-mediated decay. Indeed, Smyhc1 protein expression was not detected in smyhc1$^{-/-}$ embryos at 24 hours post fertilization (hpf) when Smyhc1 is normally most highly expressed in wild-type (smyhc1$^{+/+}$) embryos (Fig 1C–E). In contrast, smyhc1$^{R673H}$ heterozygotes and homozygotes produce detectable Smyhc1 protein, confirmed via immunohistochemistry staining (Fig 1E). The brief period of Smyhc1 protein expression is demonstrated by its absence by 48 hpf in smyhc1$^{+/+}$ larvae on Western blot (Fig 1D) and in all genotypes by 3 dpf (days post fertilization) as shown by immunohistochemistry (Fig EV1).

Consistent with prior studies that used morpholinos to transiently reduce smyhc1 expression (Codina et al, 2010), smyhc1$^{-/-}$ embryos displayed no gross phenotype in larval stages despite being paralyzed until 48 hpf (described below) (Figs 2A and 5A). In contrast, we noted that nearly half of the embryos heterozygous for the smyhc1$^{R673H}$ allele displayed mild notochord kinks or slight curves in the body axis beginning at 48 hpf, while fewer showed more severe kinks, often in multiple locations along the body axis. Furthermore, all homozygous smyhc1$^{R673H/R673H}$ zebrafish manifested severely abnormal morphology and developed multiple notochord bends and kinks that markedly compressed and disfigured the body axis and were apparent by ~ 30 hpf (Figs 2A and B, and EV3A). These early notochord kinks resemble those seen in the *accordion* zebrafish mutant, which has a muscle relaxation defect due to a mutation in the sarcoplasmic reticulum Ca$^{2+}$—ATPase pump (atp2a1) (Hirata et al, 2004).

To determine whether the presence of a wild-type copy of smyhc1 influences the phenotype of fish harboring a single smyhc1$^{R673H}$ allele, we crossed smyhc1$^{-/-}$ and smyhc1$^{R673H/+}$ mutant lines. The absence of a wild-type copy of smyhc1 resulted in more severe morphological abnormalities. In fact, the phenotype of smyhc1$^{R673H}$ transheterozygous embryos (smyhc1$^{R673H/-}$) was identical to that of smyhc1$^{R673H/R673H}$ embryos, in which similarly severe notochord kinks and bends were observed (Fig 2A). The observed partial rescue of the smyhc1$^{R673H}$ phenotype via dilution of the allele with the wild-type allele, and the failure of smyhc1$^{R673H}$ homozygotes, heterozygotes, or transheterozygotes to phenocopy smyhc1$^{-/-}$ fish support the hypothesis that the smyhc1$^{R673H}$ substitution is hypermorphic or neomorphic.

## smych1$^{R673H}$ mutation is homozygous lethal and reduces survival of heterozygotes

Because of the disruptive embryonic phenotypes of smyhc1$^{R673H}$ heterozygotes and homozygotes, we systematically determined the impact of genotype on survival. Because no smyhc1$^{R673H/R673H}$ individuals survive to adulthood for breeding, we instead incrossed smyhc1$^{R673H/+}$ fish and genotyped the resulting progeny of various ages to determine the ratio of surviving smyhc1$^{+/+}$:smyhc1$^{R673H/+}$: smyhc1$^{R673H/R673H}$ fish (Fig 3A). In contrast to the expected Mendelian ratio of 1:2:1, we discovered that smyhc1$^{R673H/R673H}$ embryos did not survive past larval stages (Fig 3B). We determined that the ratio of smyhc1$^{R673H/R673H}$ embryos sharply declined between 1 and 2 dpf, with the remaining homozygotes perishing before juvenile stages. Many smyhc1$^{R673H/+}$ fish survive into adulthood and are fertile; however, survival is reduced compared to wild type, with a

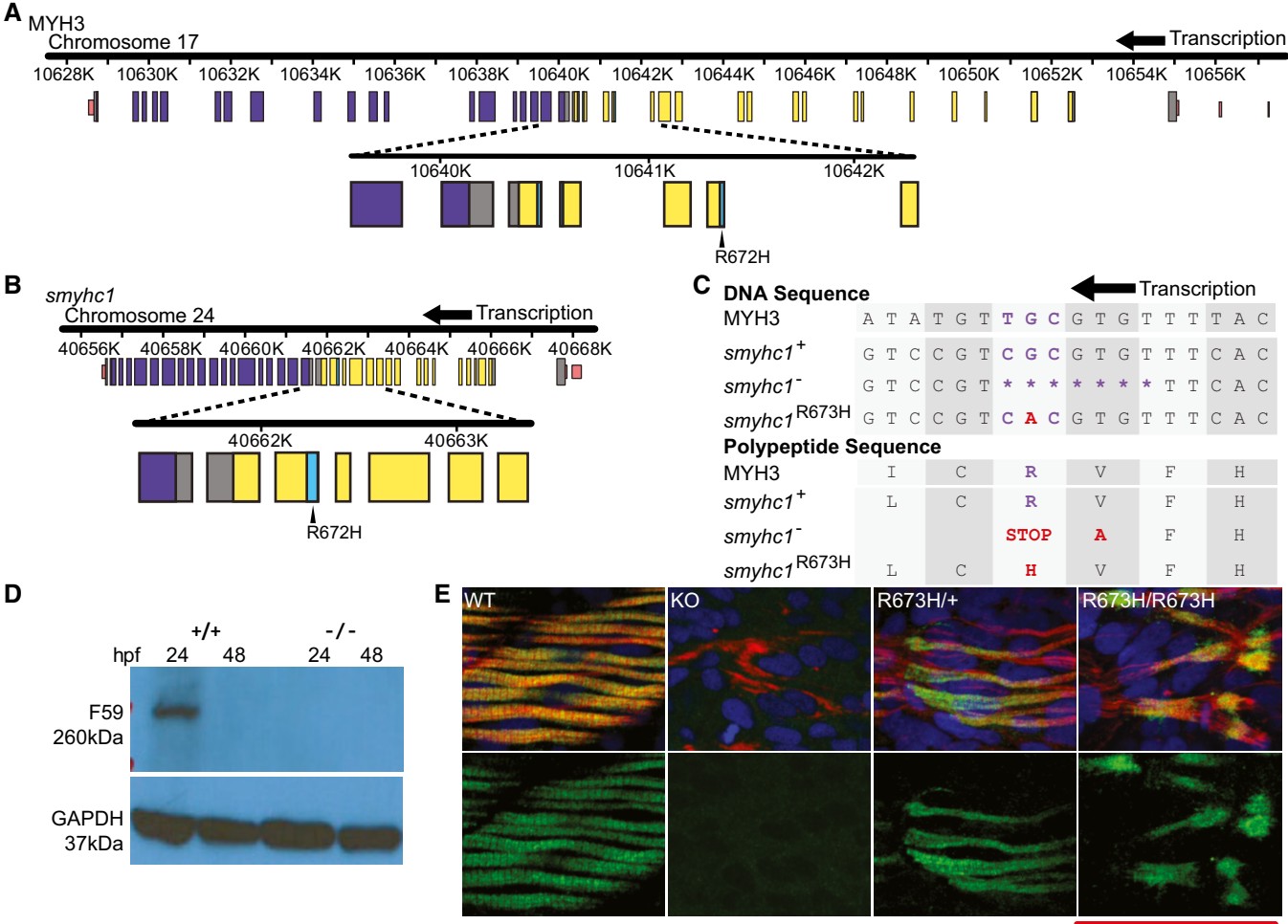

**Figure 1. Generation of *smyhc1* mutant lines and embryonic muscle development.**

A   Schematic to scale of human MYH3 gene on chromosome 17. Noncoding regions are displayed in pink. Coiled coil domain (840–1,933 bp) displayed in purple. Motor domain (86–779 bp) displayed in yellow. Actin binding site (656–678/758–772 bp) shown in cyan. Location of R672H mutation is enlarged and labeled.

B   Schematic to scale of zebrafish *smyhc1* gene on chromosome 24. Noncoding regions are displayed in pink. Coiled coil domain (842–1,929 bp) displayed in purple. Motor domain (85–778 bp) displayed in yellow. Actin binding site (655–677 bp) displayed in cyan. Location of R672H mutation is enlarged and labeled.

C   Aligned DNA and amino acid sequences of *MYH3* and *smyhc1* alleles surrounding the *smyhc1*[R673H] and *MYH3* R672H substitutions. The *smyhc1*[−] allele has a 7 base pair deletion that results in a frameshift in which one errant amino acid precedes a premature stop codon. The *smyhc1*[R673H] allele results from a G>A transition single point mutation.

D   Western blot of whole zebrafish larvae, Smyhc1 stained with the F59 antibody. *smyhc1*[+/+] larvae express Smyhc1 at 24 hpf, but not at 48 hpf. *smyhc1*[−/−] larvae do not express Smyhc1.

E   Smyhc1 immunohistochemistry stain of 24 hpf (hour postfertilization) zebrafish larvae. Filamentous actin is stained with phalloidin (red). Nuclei are stained with DAPI (blue). Smyhc1 is stained with the F59 antibody (green) (Elworthy *et al*, 2008). *smyhc1*[+/+], *smyhc1*[R673H/+], and *smyhc1*[R673H/R673H] larvae display Smyhc1 in the muscle fibers at 24 hpf. *smyhc1*[−/−] larvae at 24 do not stain positively for Smyhc1. Scale bar represents 50 μm.

decline continuing throughout adulthood. The *smyhc1*[R673H] allele increases mortality of the fish in a dose-dependent manner, consistent with the severity of the gross phenotype.

### *smyhc1*[−/−] adults develop late-onset spinal curvature

We noticed an increase in spinal curvature of *smyhc1*[−/−] fish as they reached adulthood, and decided to examine the phenotype further. Despite the apparent short and early temporal expression of *smyhc1*, which is strongly expressed at 24 hpf but markedly reduced by 48 hpf (Fig 1D), its disruption has lasting effects into adulthood.

Although *smyhc1*[−/−] larvae appeared morphologically normal (Fig 2A), nearly all developed spinal curvatures in adulthood (Figs 4A and EV2). To determine whether the observed curvature is associated with vertebral fusions, alizarin red/alcian blue staining of bone and cartilage, respectively, was performed on adult zebrafish between 1 and 2 years old. Alizarin red staining of adult *smyhc1*[−/−] fish revealed variable spinal curvatures along the length of the spine in both lateral and sagittal planes (Fig 4B). Most *smyhc1*[−/−] fish were afflicted with mild-to-moderate curvature (Figs 4C and EV3B). Despite the apparent curvature of *smyhc1*[−/−] fish, vertebral fusions were not observed. This is consistent with the lack of notochord

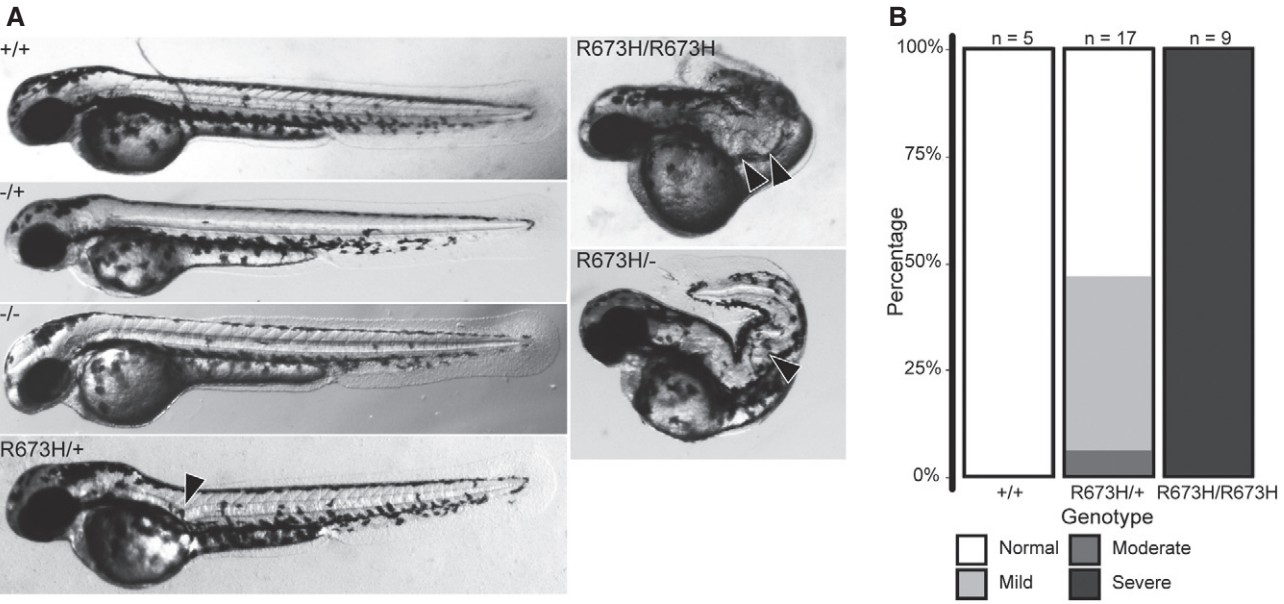

**Figure 2. Embryonic phenotypes of *smyhc1* mutants.**

A   Morphologies of *smyhc1*$^{+/+}$, *smyhc1*$^{-/+}$, and *smyhc1*$^{-/-}$ embryos are grossly normal at 2 days postfertilization (dpf), despite complete paralysis of smyhc1$^{-/-}$ embryos. In contrast, the *smyhc1*$^{R673H/+}$ embryos have notochord kinks in larval stages (arrow head), while smyhc1$^{R673H/R673H}$ embryos have much more severe notochord kinks that compress and distort the body axis (arrow heads). The smyhc1$^{R673H/-}$ embryos completely phenocopy *smyhc1*$^{R673H/R673H}$ embryos.

B   Quantification of *smyhc1*$^{R673H}$ embryonic phenotypes at 2 dpf. Representative examples of fish in each of the four phenotypic groups (normal, mild, moderate, and severe) are shown in Fig EV3.

abnormalities displayed in larval stages in *smyhc1*$^{-/-}$ fish, a phenotype associated with later vertebral fusions in other mutant fish lines (reviewed in Ellis *et al*, 2013a,b; Gray *et al*, 2014).

## *smyhc1*$^{R673H/+}$ adults have both spinal curvature and vertebral fusions

Similar to the *smyhc1*$^{-/-}$ fish, the *smyhc1*$^{R673H/+}$ fish displayed spinal curvatures in adulthood, but the fish were often shortened in length (Figs 4A and EV2). In addition, the spinal curves developed much earlier; notochord kinks and bends that were seen as early as 48 hpf often preceded the later development of structural vertebral abnormalities (Figs 2A and 4B), much as those previously reported in *leviathan/col8a1a* zebrafish mutants (Gray *et al*, 2014). Upon observation of alizarin red stained bone, most *smyhc1*$^{R673H/+}$ adults were found to have moderate-to-severe vertebral abnormalities, with complex and variable vertebral fusions often involving multiple vertebrae and their jaws often had a closed mouth appearance (Figs 4B and C, and EV3B). Similar to leviathan, the vertebral fusions in *smyhc1*$^{R673H/+}$ were present at regions of severe spinal curve and were not detected in regions where the spine was straight. Spinal curves were predominantly present in the lateral plane and were nearly all restricted to the distal tail. However, we noted that some *smyhc1*$^{R673H/+}$ larvae with proximal notochord kinks did not survive to adulthood, which suggests that the propensity toward distal spinal curves may be due to a selection bias rather than a developmental predisposition to distal scoliosis. These data suggest that larval notochord abnormalities predispose *smyhc1*$^{R673H/+}$ fish to vertebral fusions in adulthood.

## *smyhc1* mutants have motor deficits

Because *smyhc1* is a myosin heavy chain gene critical for motor function, and because of the observed effects of *smyhc1* mutations on gross anatomy described above, we assessed the effects of *smyhc1* mutations on muscle function by quantifying the movement of fish at various times during development. The *smyhc1* gene is expressed embryonically, as early as 10-13 somites as evidenced by *in situ* hybridization (Rauch *et al*, 2003; Bessarab *et al*, 2008; Li *et al*, 2020); therefore, we first assessed embryonic movements by counting the number of light-induced twitches beginning at 24 hpf when embryos first respond to light stimulation (Kokel & Peterson, 2011; Saint-Amant & Drapeau, 1998; Fig 5A). The *smyhc1*$^{-/-}$ embryos were completely paralyzed up to 48 hpf, after which they exhibited normal movements in response to light stimulation. There was not a statistically significant difference between *smyhc1*$^{R673H/+}$ and their wild-type siblings until 34 hpf, when mutants displayed a slight decrease in the number of twitches (Fig 5A). At early stages when muscle function was grossly measured as the ability to twitch in response to light, the *smyhc1*$^{R673H}$ allele had only minimally apparent effects while lack of Smyhc1 abolished muscle function in early embryonic stages.

Motor function was also assessed at 6 dpf to determine later effects of *smyhc1* mutations (Fig 5B). Automated motion tracking software was used to quantitatively measure spontaneous swimming distance, which was not possible at earlier stages due to their small size and transparency. The total distance traveled over 5 minutes was less in *smyhc1*$^{R673H/+}$ larvae compared to their wild-type siblings. The *smyhc1*$^{-/-}$ larvae swam a significantly shorter

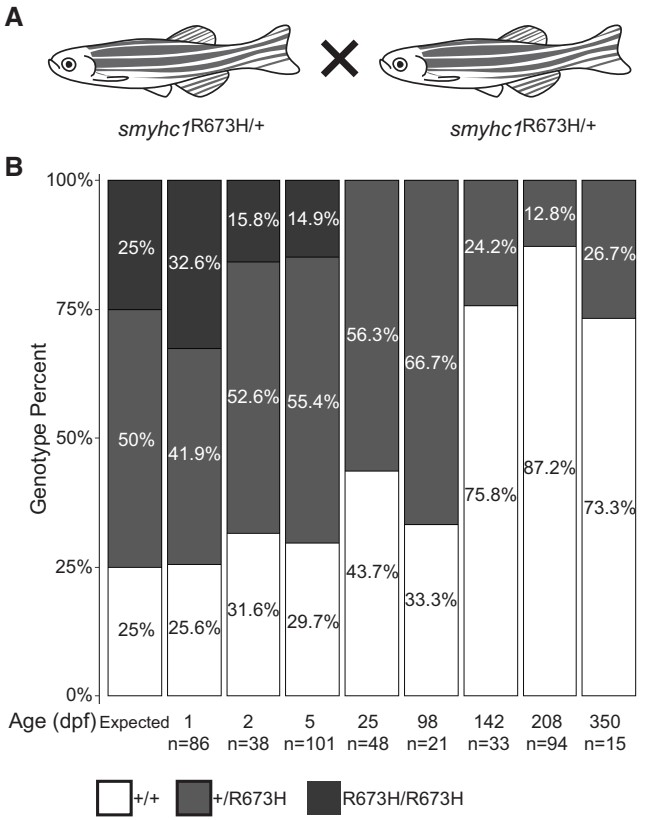

**A**

smyhc1R673H/+    ×    smyhc1R673H/+

**B**

**Figure 3. The smyhc1R673H allele reduces survival in a dose-dependent manner.**

A  Schema of in-crosses used to generate clutches for survival assay.
B  The predicted mendelian (1:2:1) and experimental ratio of genotypes of progeny from smyhc1R673H/+ in-crosses demonstrates survival at the indicated ages. This ratio quickly skewed toward an overrepresentation of smyhc1+/+ fish with no representation of smyhc1R673H/R673H after 5 dpf. There is also dropout of smyhc1R673H/+ fish into adulthood. Number of fish counted per group is indicated.

distance than both of the other genotypes (Fig 5B). At larval stages, it appears that the effects of smyhc1 mutations, even after apparent downregulation of the gene, have an enduring effect on muscle function and locomotion.

Muscle function was measured in adults at 6 months postfertilization (mpf) by assessing endurance upon exposure to a gradually escalating flow of water in a swim tunnel (Fig 5C). The smyhc1R673H/+ fish fatigued in the swim tunnel sooner than smyhc1+/+ and smyhc1−/− fish. No significant difference was noted in time spent swimming between the smyhc1+/+ and smyhc1−/− fish (Fig 5D). While the smyhc1− allele had a more severe effect on muscle function in larval stages compared to the smyhc1R673H allele, this result provides evidence that smyhc1−/− mutants recover a normal degree of motor function in adulthood.

## smyhc1 R673H mutants have disorganized muscle and shortened myoseptal intervals

Because smyhc1 is the first and earliest myosin heavy chain expressed in slow skeletal muscle (Devoto et al, 1996), we examined the effect

of mutant smyhc1 alleles on morphological muscle development. Phalloidin-rhodamine was used to stain filamentous actin in smyhc1+/+, smyhc1−/−, smyhc1R673H/+, and smyhc1R673H/R673H larvae. At 24 hpf, the smyhc1−/− muscle was highly disorganized and had no distinguishable sarcomeres or myofibers (Fig 6A and C). Wavy strands of filamentous actin and numerous puncta of actin bundles were present (Fig 6A). In contrast, smyhc1R673H heterozygotes and homozygotes formed sarcomeres, but the filamentous actin was unevenly distributed across the myofiber, which occasionally had a flared appearance near the myosepta (Fig 6A). At 3 dpf, discrete sarcomeres became evident in smyhc1−/− myofibers; however, they were poorly aligned with each other and with the myosepta. smyhc1−/− larvae were also found to have centralized nuclei (Fig 6B and D). Organized sarcomeres were also present in smyhc1R673H/+ muscle, but myofibers were thin, becoming bulbous at their termini, leading to poorly defined myosepta. The smyhc1R673H/R673H myofibers were also had poorly defined myoseptal boundaries; bundles of actin were commonly observed (Fig 6B and D).

To better define the role of smyhc1 in sarcomere development, Z-disks were labeled with anti-α-actinin antibodies. At 24 hpf, Z-disks were assembled in smyhc1+/+, smyhc1R673H/+, and smyhc1R673H/R673H larvae, while smyhc1−/− larvae failed to develop any Z-disk structure (Fig 6C). However, by 3 dpf, smyhc1−/− larvae formed Z-disk structures, with a disorganized appearance consistent with the thin filament morphology (Fig 6D). We then quantified the distance between Z-disks to determine whether sarcomere length was affected by the R673H mutation or lack of Smyhc1. smyhc1R673H/R673H and smyhc1R673H/+ genotypes displayed significantly shorter sarcomere lengths compared to wild type at both 24 hpf and 3 dpf, with a large range of sarcomere length variability in smyhc1R673H/R673H larvae (Fig 6G and H). At 3 dpf, smyhc1−/− fish displayed a significantly shorter Z-disk interval than any other genotype.

To determine whether the smyhc1 alleles affected the overall somite length as was previously observed in the hypercontractile accordion mutant (Hirata et al, 2004), we measured the distance between myosepta in slow skeletal muscle. Myoseptal intervals were significantly reduced in smyhc1R673H/R673H and smyhc1R673H/+ muscle at 24 hpf and 3 dpf. smyhc1−/− larvae did not display a shortened myoseptal distance (Fig 6E and F). Because both the Z-disk interval and myoseptal interval are reduced in smyhc1R673H mutants, we hypothesize that the shortened myoseptal interval results from increased tension shortening the length of the sarcomere, and subsequently the myofiber, as was proposed for the accordion mutant (Hirata et al, 2004). Conversely, the shortened Z-disk interval observed in smyhc1−/− fish is not accompanied by a shortened myoseptal distance. These data, in addition to the observed disorganization of sarcomere alignment in smyhc1−/− fish, suggest a different mechanism, perhaps in which the structural effects caused by the lack of Smyhc1 result in both a shortened Z-disk interval and sarcomere misalignment.

## Cell death in smyhc1R673H/R673H zebrafish

To explore the effects of the smyhc1 alleles on cell viability, larvae at 24 hpf and 3 dpf were stained with Caspase-3 antibodies (Dalgin et al, 2011; Dalgin & Prince, 2015). Increased signal of the cleaved Caspase-3 protein indicates an activation of the apoptosis pathway (Sorrells et al, 2013). While there was no difference in cleaved Caspase-3 at 24hpf, smyhc1R673H/R673H larvae had increased cleaved

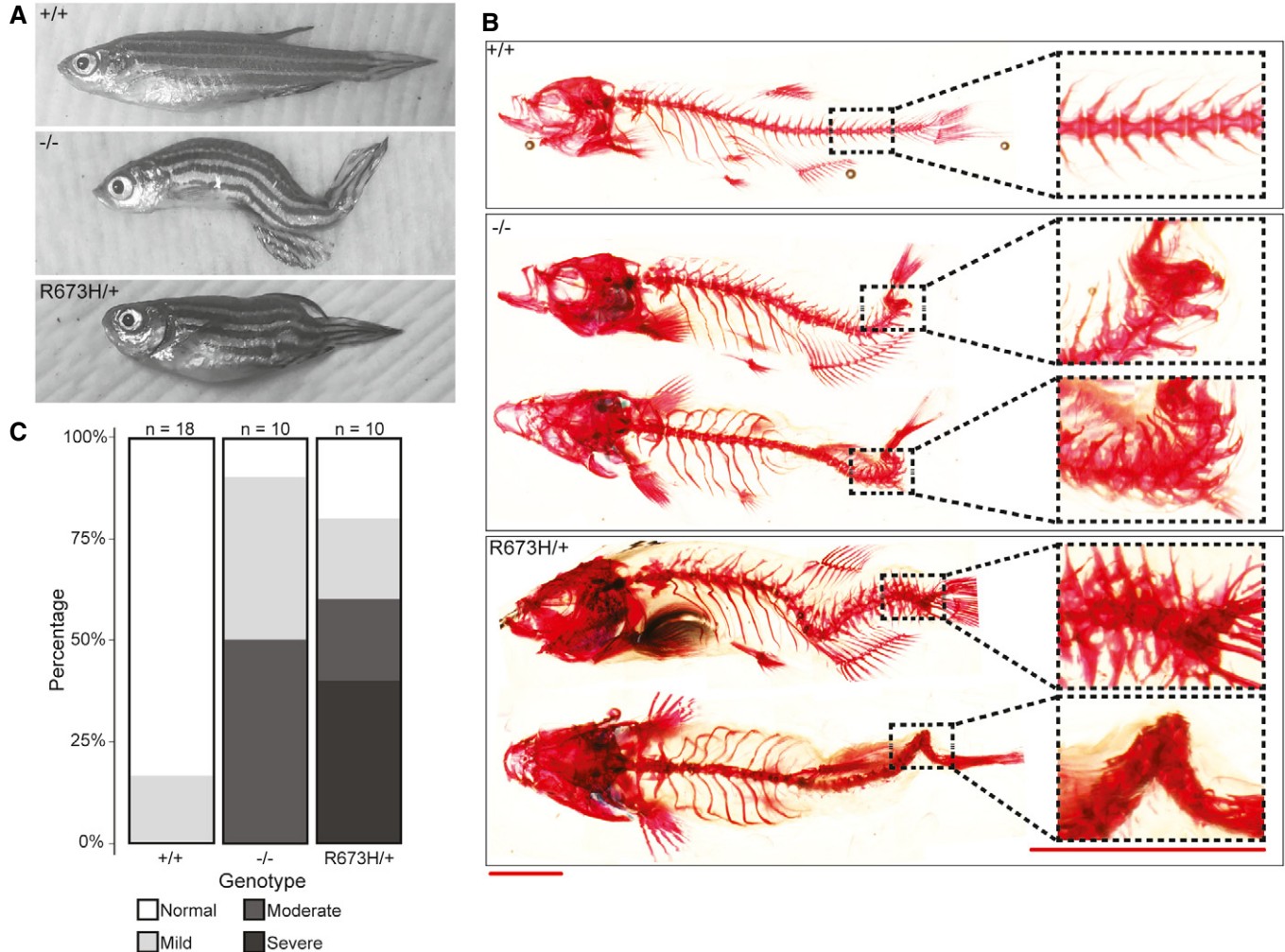

**Figure 4. *smyhc1* mutants exhibit skeletal abnormalities in adulthood.**

A  Gross morphology of *smyhc1* mutant adults. Most *smyhc1*[−/−] adult fish display dorsal tail curvature, while *smyhc1*[R673H/+] adults have a shortened body axis and variable spinal curves.

B  Alizarin red staining of bone shows spinal curvature but no bony fusions in the skeleton of *smyhc1*[−/−] adults, in contrast to the compression and fusion of vertebrae seen in *smyhc1*[R673H/+] adults. Distal tail regions are highlighted and enlarged.

C  Quantification of skeletal phenotype severity in adults. Representative examples of fish in each of the four phenotypic groups (normal, mild, moderate, and severe) are shown in Fig EV3. Scale bars represent a length of 5 mm.

Caspase-3 in the muscle tissue at 3dpf (Fig EV4). This suggests that increased tissue stress and muscle cell morphological decline due to the *smyhc1*[R673H] allele may lead to an increase in cell death in the homozygous state.

**Para-aminoblebbistatin rescues *smyhc1*[R673H] mutant phenotypes**

To understand the proposed mechanism and explore new therapeutic approaches to prevent contractures and disability in DA patients, we evaluated the ability of select pharmacological agents to normalize the phenotype of *smyhc1*[R673H] heterozygous and homozygous embryos. Tricaine methanesulfonate, an anesthetic that acts as a neuromuscular blocking agent by directly inhibiting neuronal sodium channels (Attili & Hughes, 2014), was applied to dechorionated embryos from 12 to 72 hpf. Although the drug completely

suppressed embryonic movement as expected, it failed to improve the ventral body curvature or reduce the development of notochord kinks in *smyhc1*[R673H] heterozygotes and homozygotes (Fig EV5A). This suggests that the contractures observed in *smyhc1*[R673H] mutants arise independently of stimulated muscle contraction, but is alternatively intrinsic to the muscle, and present even without neural stimulation.

Myosin ATPase inhibitors are being developed for the treatment of cardiomyopathy caused by similar mutations of cardiac myosin heavy chain genes (Green *et al*, 2016); therefore, we sought to determine whether drugs that directly inhibit function of myosin (Kovács *et al*, 2004) could reduce the adverse phenotypic effects of the *smyhc1*[R673H] allele. Blebbistatin, a myosin ATPase inhibitor, which reduces actin–myosin affinity, was applied to dechorionated embryos at a concentration of 25 μM

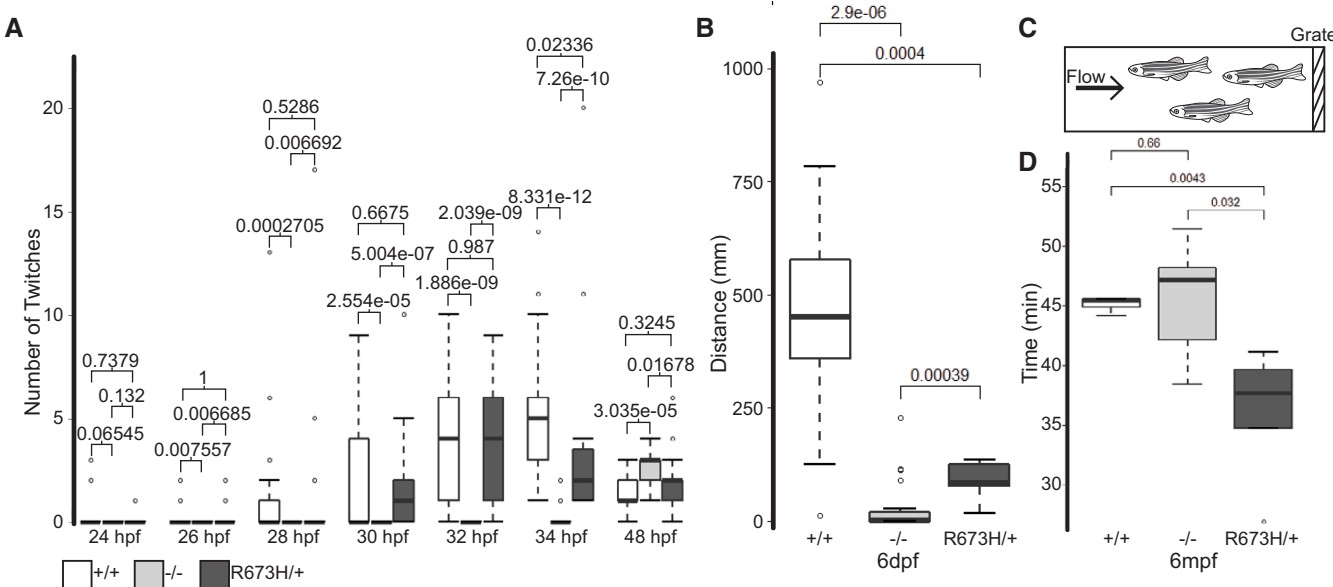

**Figure 5. Motor deficits in *smyhc1* mutants are variable during development.**

A   Time course of light-triggered larval movements was manually counted at indicated times between 24 and 48 hpf. The *smyhc1*[−/−] larvae (*n* = 36) were paralyzed up to 48 hpf. *smyhc1*[R673H/+] mutants (*n* = 15) twitched slightly less than their *smyhc1*[+/+] siblings (*n* = 21) at 34 hpf. The central bands of the boxplots represent the median, the boxes of the boxplots represent the interquartile range (between the first and third quartile), and the whiskers represent the minimum and maximum values, up to 1.5 times the interquartile range. Outliers displayed are outside of this range.

B   Distance traveled during 5 min of spontaneous swimming at 6 dpf quantified by motion tracking software (Noldus Ethovision). At 6 dpf, both *smyhc1*[−/−] (*n* = 24) and *smyhc1*[R673H/+] (*n* = 11) larvae traveled significantly less than *smyhc1*[+/+] larvae (*n* = 13). The central bands of the boxplots represent the median, the boxes of the boxplots represent the interquartile range (between the first and third quartile), and the whiskers represent the minimum and maximum values, up to 1.5 times the interquartile range. Outliers displayed are outside of this range.

C   Diagram of swim tunnel used to quantify adult swimming behavior. Water flow was gradually increased until the fish fatigued and were collected at the grate.

D   Time spent swimming in swim tunnel before fatigue in adults at 6 months postfertilization (mpf). *smyhc1*[+/+] (*n* = 6), *smyhc1*[−/−] (*n* = 5), *smyhc1*[R673H/+] (*n* = 5).

Data Information: Wilcoxon rank-sum test used to calculate significance. *P*-values comparing each group displayed above data. The central bands of the boxplots represent the median, the boxes of the boxplots represent the interquartile range (between the first and third quartile), and the whiskers represent the minimum and maximum values, up to 1.5 times the interquartile range. Outliers displayed are outside of this range.

from 24 to 48 hpf (Wang *et al*, 2015). Blebbistatin prevented the development of notochord kinks in both *smyhc1*[R673H/+] and *smyhc1*[R673H/R673H] embryos (Rauscher *et al*, 2018; Fig EV5B). However, numerous developmental defects, including pericardial edema, dorsal tail curvature, and small eyes, were observed in all fish exposed to this drug, including wild-type larvae, consistent with the known cytotoxic effects of blebbistatin that are independent of its myosin inhibition (Várkuti *et al*, 2016).

A blebbistatin derivative, para-aminoblebbistatin, which is more photostable and less cytotoxic (Várkuti *et al*, 2016), was therefore applied to dechorionated embryos at 25 μM for 24 or 48 h, starting at 24 hpf. While para-aminoblebbistatin does not share the cytotoxic effects of blebbistatin, its effect on cardiac myosin appears to cause lethal pericardial edema in both mutant and wild-type embryos (Várkuti *et al*, 2016). However, similarly to blebbistatin, its effect on skeletal myosin completely abolished notochord kinks and bends in both *smyhc1*[R673H/+] and *smyhc1*[R673H/R673H] embryos at 2 dpf (Fig 7A). This effect persisted to 3 dpf (Fig 7B). The molecular inhibition of actin–myosin interaction removes the ability of the muscle to contract. This effect suppresses the *smyhc1*[R673H] phenotype, suggesting that the allele causes notochord kinking as a secondary effect of constitutive myofiber contraction, independent of neural stimulation.

# Discussion

Understanding the functional consequences of *MYH3* variants associated with human disease has been limited by the lack of access to human skeletal muscle tissue during embryonic development. The recent description of multiple human skeletal muscle and bone phenotypes associated with *MYH3* variants, including autosomal dominant and recessive spondylocarpotarsal synostosis syndromes, in addition to DA, provides a strong motivation to develop animal models to study and understand these variable phenotypes. By generating two viable zebrafish *smyhc1* alleles, including a *smyhc1* null allele and a *smyhc1*[R673H] missense variant, we now have tools to assess inheritance models and effects of gene dosage on phenotypic expression.

Our results not only confirm that the *smyhc1*[R673H] allele is more severe than a *smyhc1* null allele (which was also recently described in Li *et al*, 2020), but also demonstrate the dosage sensitivity of the missense allele to the wild-type allele, which modulates the effects of the disease-causing R673H variant in our model system. Expression of the *smyhc1*[R673H] allele in the absence of a wild-type allele resulted in embryos with a lethal phenotype identical to *smyhc1*[R673H/R673H] embryos. Myosin is comprised of a hexamer containing a pair of heavy chains, a pair of regulatory light chains,

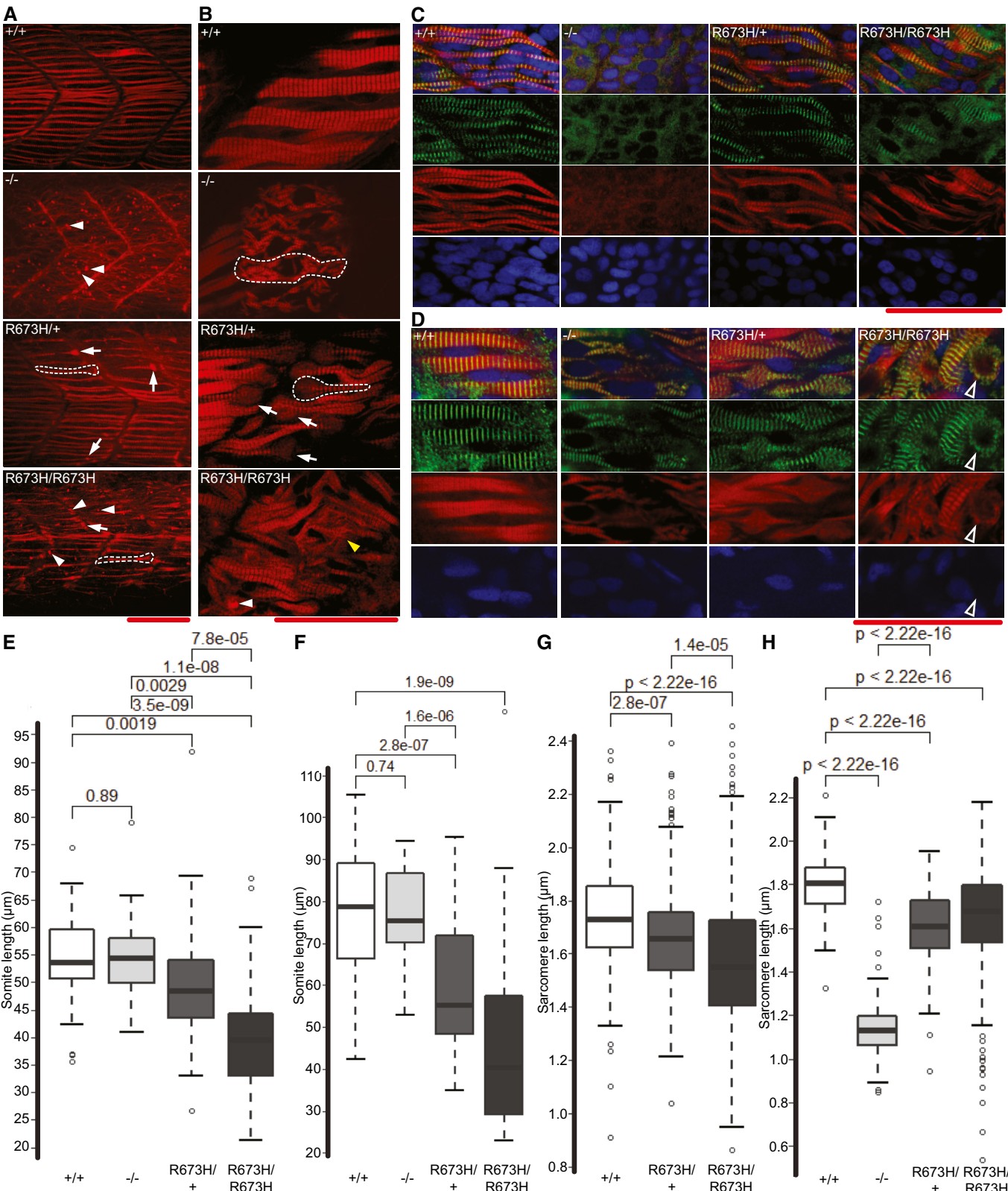

**Figure 6.**

Figure 6. Slow skeletal muscle disorganization of *smyhc1* mutants.

A  Confocal fluorescence images of slow skeletal muscle of 1 dpf larvae, stained with phalloidin-rhodamine to detect actin. Puncta of filamentous actin are indicated with white arrowheads, and misshapen myofibers are indicated with white arrows and outlined.

B  Confocal microscope fluorescence images of slow skeletal muscle of 3 dpf larvae, stained with phalloidin-rhodamine. Puncta of filamentous actin are indicated with white arrowheads, distorted myofibers are indicated with arrows and outlined, and frayed myofibers are indicated with yellow arrowhead.

C  Confocal fluorescence images of slow skeletal muscle of 1 dpf larvae, stained with phalloidin-rhodamine (red), anti-α-actinin antibodies (green), and DAPI (blue).

D  Confocal fluorescence images of slow skeletal muscle of 3 dpf larvae, stained with phalloidin-rhodamine (red), anti-α-actinin antibodies (green), and DAPI (blue). Bundle of actin ringed with α-actinin indicated with white-bordered black arrowhead.

E  Myoseptal intervals of slow skeletal muscle at 1 dpf in $smyhc1^{+/+}$ (n = 54), $smyhc1^{-/-}$ (n = 31), $smyhc1^{R673H/+}$ (n = 45), and $smyhc1^{R673H/R673H}$ (n = 49). Distance between myosepta was measured perpendicular to rostral-caudal body axis at defined mid-body regions in phalloidin stained slow skeletal muscle fluorescence images. The central bands of the boxplots represent the median, the boxes of the boxplots represent the interquartile range (between the first and third quartile), and the whiskers represent the minimum and maximum values, up to 1.5 times the interquartile range. Outliers displayed are outside of this range.

F  Myoseptal intervals of slow skeletal muscle at 3 dpf in $smyhc1^{+/+}$ (n = 51), $smyhc1^{-/-}$ (n = 32), $smyhc1^{R673H/+}$ (n = 64), and $smyhc1^{R673H/R673H}$ (n = 38). The central bands of the boxplots represent the median, the boxes of the boxplots represent the interquartile range (between the first and third quartile), and the whiskers represent the minimum and maximum values, up to 1.5 times the interquartile range. Outliers displayed are outside of this range.

G  Z-disk intervals (sarcomere length) of slow skeletal muscle at 1 dpf in $smyhc1^{+/+}$ (n = 380), $smyhc1^{R673H/+}$ (n = 293), and $smyhc1^{R673H/R673H}$ (n = 268). Distance between z-disks was measured in anti-α-actinin stained slow skeletal muscle fluorescence images. The central bands of the boxplots represent the median, the boxes of the boxplots represent the interquartile range (between the first and third quartile), and the whiskers represent the minimum and maximum values, up to 1.5 times the interquartile range. Outliers displayed are outside of this range.

H  Z-disk intervals (sarcomere length) of slow skeletal muscle at 3 dpf in $smyhc1^{+/+}$ (n = 249), $smyhc1^{-/-}$ (n = 278), $smyhc1^{R673H/+}$ (n = 246), and $smyhc1^{R673H/R673H}$ (n = 252). The central bands of the boxplots represent the median, the boxes of the boxplots represent the interquartile range (between the first and third quartile), and the whiskers represent the minimum and maximum values, up to 1.5 times the interquartile range. Outliers displayed are outside of this range.

Data Information: Wilcoxon rank-sum test used to calculate significance. *P*-values comparing each group displayed above data. Scale bars represent a length of 50 μm.

and a pair of essential light chains (Schiaffino & Reggiani, 1996). Because of the dimerization of the myosin heavy chains, and subsequent assembly of the hexamers into the thick filament, it is reasonable to hypothesize that compound heterozygosity of a missense mutation in trans with a hypomorphic or null allele will increase the phenotypic severity of afflicted individuals compared to a missense mutation in trans with a wild-type allele. In fact, a Dutch study recently identified a high impact *MYH3* 5′ UTR variant with a gnomAD minor allele frequency of 0.008 that diminishes *MYH3* translational efficiency (Cameron-Christie *et al*, 2018). This variant was reported in patients with presumed autosomal recessive spondylocarpotarsal synostosis syndrome, demonstrating a biallelic phenotype consisting of a haploinsufficient allele and a hypomorphic allele. This genotype exacerbated the severity of the disorder, and our results in zebrafish similarly demonstrate that the phenotypic severity of an autosomal dominant allele is greater when in trans with a hypomorphic than a wild-type allele. Because intrafamilial phenotypic variability has previously been reported in DA1 families with *MYH3* mutations (Alvarado *et al*, 2011), we now hypothesize that expression differences of the wild-type allele may be responsible.

Zebrafish offer many practical advantages for research, including accessibility at early time points in development due to their optical clarity, speed of development, large clutch size, and convenient external development. *smyhc1* is the only myosin gene expressed during early zebrafish embryogenesis; therefore, we hypothesized that its manipulation would accurately model the human condition (Devoto *et al*, 1996; Elworthy *et al*, 2008; Schiaffino *et al*, 2015). Zebrafish also offer a unique opportunity to study slow skeletal muscle development in isolation due to the spatial segregation of slow and fast skeletal muscle in zebrafish, with the former developing as a thin border along the lateral aspect and the latter making up the bulk of the trunk (Elworthy *et al*, 2008). In contrast, human skeletal muscle is composed of interspersed slow and fast twitch muscle fibers. Muscle hypercontractility in *smyhc1*^R673H heterozygous and homozygous fish was evidenced by the shortened slow

muscle sarcomere length, and shortened myoseptal interval that phenocopies the *accordion* hypercontractile zebrafish mutant which has a muscle relaxation defect due to a mutation in *atp2a1* (sarcoplasmic reticulum Ca²⁺-ATPase 1; Hirata *et al*, 2004). In our model, some myofibers even appeared to physically dissociate from the myosepta by flaring bulbously at the end of the myofiber, which we hypothesize was a consequence of excessive biomechanical force and may have contributed to increased apoptosis in the homozygous *smyhc1*^R673H larvae. This resembles the hypercontractile phenotype seen in *Drosophila* upon expression of the R672C mutation, in which normal myofibril assembly is followed by degeneration that ultimately results in a complete impairment of flight (Rao *et al*, 2019).

One of the earliest morphological abnormalities that we observed in *smyhc1*^R673H heterozygous and homozygous mutant embryos was prominent notochord kinks and bends. Results from our study suggest that notochord abnormalities arise as a consequence of excessive tension on the notochord due to hypercontraction and/or uncoordinated contraction of the surrounding muscle. Data to support this mechanism include the much more severe phenotype seen in homozygous *smyhc1*^R673H mutants, and the complete rescue of this phenotype by treatment with the myosin ATPase inhibitor, para-aminoblebbistatin. While the notochord abnormalities in the *accordion* mutant were suppressed with drugs inhibiting neural transmission (Hirata *et al*, 2004), the *smyhc1*^R673H mutant notochord defects could not be similarly suppressed, but improved only when muscle contraction itself was directly impaired via myosin-inhibiting drugs. Our *in vivo* results are consistent with prior *in vitro* studies of both patient muscle tissue and C2C12 cells in which the R673H variant was observed to negatively affect relaxation kinetics (Racca *et al*, 2015; Walklate *et al*, 2016). As predicted by their work, gross phenotypic abnormalities in our *smyhc1*^R673H mutants, including severe notochord kinks and bends, were completely abolished after treatment with myosin inhibitors.

The phenotypic spectrum of *MYH3*-associated disease was recently broadened to include vertebral fusions, which are a

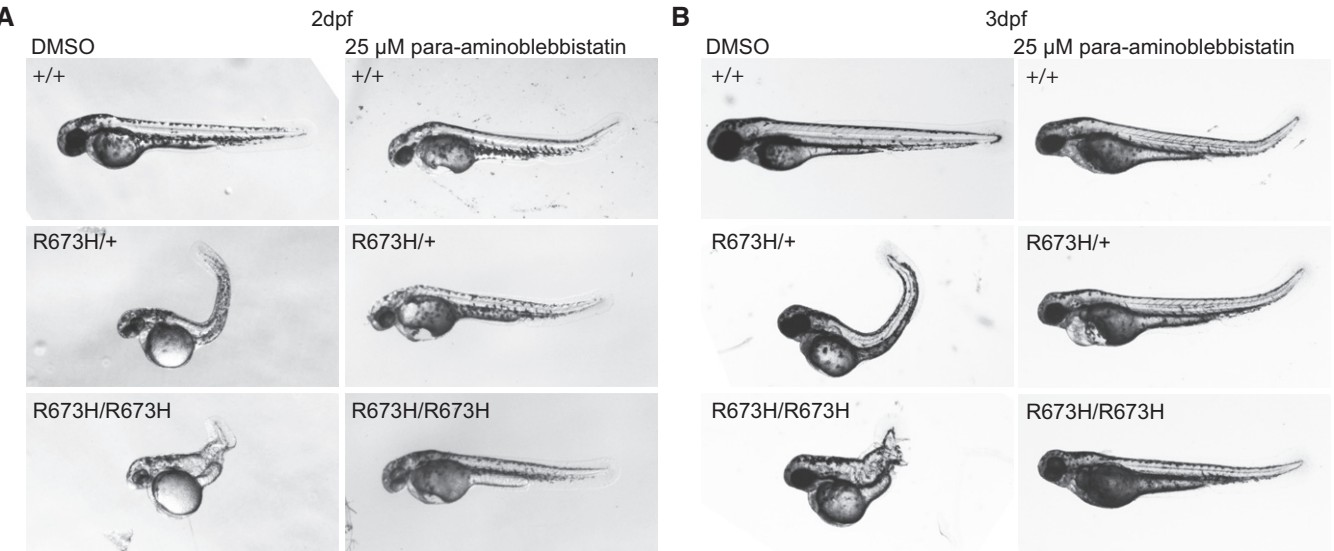

**Figure 7. Para-aminoblebbistatin ameliorates the smyhc1^R673H larval phenotype.**

A Representative morphologies of *smyhc1*^+/+, *smyhc1*^R673H/+, and *smyhc1*^R673H/R673H larvae at 2 dpf after 24-h treatment with 0.25% DMSO (control) or 25 µM para-aminoblebbistatin. All larvae treated with para-aminoblebbistatin develop a slight dorsal tail curve and pericardial edema regardless of genotype. All para-aminoblebbistatin-treated *smyhc1*^R673H/+ (*n* = 9) and *smyhc1*^R673H/R673H (*n* = 3) larvae are indistinguishable from *smyhc1*^+/+ unlike the untreated larvae which have markedly curved or distorted body axis.

B Representative images of *smyhc1*^+/+, *smyhc1*^R673H/+, and *smyhc1*^R673H/R673H larvae at 3 dpf after 48-h treatment with 0.25% DMSO (control) or 25 µM para-aminoblebbistatin. All larvae treated with para-aminoblebbistatin develop a slight dorsal tail curve and pericardial edema regardless of genotype. Para-aminoblebbistatin-treated *smyhc1*^R673H/+ (*n* = 9) and *smyhc1*^R673H/R673H (*n* = 3) larvae are indistinguishable from *smyhc1*^+/+ unlike the untreated larvae which have markedly curved or distorted body axis.

defining clinical feature of spondylocarpotarsal synostosis syndrome. Spondylocarpotarsal synostosis syndrome is associated with both autosomal dominant and recessive inheritance; both missense and nonsense *MYH3* variants have been reported in recessive cases (Carapito *et al*, 2016; Zieba *et al*, 2017; Cameron-Christie *et al*, 2018). Congenital spinal fusions were observed in our *smyhc1*^R673H mutants but not in *smyhc1* null mutant fish, suggesting that bony fusions in our missense model result secondarily from increased tension generated by the mutant muscle protein, putting excessive tension and stress on the notochord, causing it to kink. Gray *et al* (2014) previously observed a direct spatial relationship between the areas where notochord kinks were most severe and the location of later developing vertebral fusions in *leviathan*/*col8a1a* null zebra-fish. Thus, our current model supports a hypermorphic or neomorphic effect of the *smyhc1*^R673H allele that causes increased muscle tension, which in turn puts excessive stress on the notochord resulting in kinks that progress into vertebral fusions in adult fish. Others have proposed alternative mechanisms for the development of skeletal abnormalities in *MYH3*-associated disorders, including a direct effect of its postnatal expression in vertebral bone (Zieba *et al*, 2017). Furthermore, the postnatal persistence of *MYH3* expression in the small multifidus muscles connecting to the neural arches of the spine in mice was also proposed as a possible mechanism. It is possible that abnormal mechanical forces imposed upon the intervertebral disk by pathogenic *MYH3* mutations functioning postnatally in these small muscle increase the vulnerability of the spine to scoliosis and spinal fusions over time (Zieba *et al*, 2017). Furthermore, in cultured cells, *MYH3* expression also activates TGF-β

signaling pathways that are associated with bone fusions in other disease models (Zieba *et al*, 2017). Interestingly, we also note that *smyhc1* is expressed in zebrafish somites at the 5–9 somite stage even prior to muscle development (Rauch *et al*, 2003; Bessarab *et al*, 2008; Li *et al*, 2020); its role at this early time period is unknown but may indicate a function independent from its role in muscle. More work is needed to understand the how *MYH3* mediates the complex developmental relationship between muscle and bone.

Animal models are necessary to advance the development of novel gene-directed therapies for patients with DA. Pharmacological approaches for DA should be informed by research in hypertrophic cardiomyopathy on the closely related cardiac myosin heavy chain gene, *MYH7* (Green *et al*, 2016). The R453 residue in *MYH7* is analogous to the R673 residue targeted for mutation here in the zebrafish gene, *smyhc1*. Prior work has modeled the *MYH7* R453C variant in mice, demonstrating that treatment with cardiac specific myosin inhibitors, including the small molecule MYK-461, reduces ventricular hypertrophy, cardiomyocyte disarray, and profibrotic gene expression that are hallmarks of human hypertrophic cardiomyopathy (Green *et al*, 2016). While inhibitors specific for embryonic skeletal muscle myosin have not yet been reported, they would be ideal drugs for the treatment and prevention of diseases associated with *MYH3* hypermorphic mutations. If *MYH3*-associated muscle contractures or bone fusions result entirely from early embryonic gene expression, then short drug treatments may be sufficient to prevent progression of pathogenic phenotypes. Although our data show that zebrafish with complete loss of *smyhc1* expression have a

milder phenotype than those with $smyhc1^{R673H}$ mutations, nonspecific inhibition of *smyhc1* or *MYH3* with antisense oligonucleotides risks worsening the phenotype of patients with missense mutations, as seen by the severe phenotype of $smyhc1^{R673H/-}$ mutants. Therefore, any proposed antisense oligonucleotide therapy would need to be designed specifically for each *MYH3* pathogenic variant. Our work supports the potential use of myosin ATPase inhibitors for the treatment of *MYH3*-associated DA, but considerable challenges remain, including elucidation of the optimal treatment period and reduction of toxicity through the development of embryonic myosin specific inhibitors.

# Materials and Methods

### Ethical statement

Husbandry and protocols for experiments involving zebrafish (*Danio rerio*) have been reviewed and approved by the Institutional Animal Care and Use Committee (IACUC) at Washington University.

### Zebrafish husbandry

Zebrafish (*Danio rerio*) were raised and maintained using standard methods (Westerfield, 1995). Zebrafish were housed and handled using protocols approved by the Institutional Animal Care and Use Committee (IACUC) at Washington University.

### TALEN (transcription activator-like effector nuclease) generation of *smyhc1* mutants

Left and right TALENs were designed to target the sites flanking codon 673 of *smyhc1*, on exon 16. TALENs were generated using the Golden Gate method as previously described (Cermak *et al*, 2011). Plasmids were acquired from the Golden Gate TALEN and TAL Effector Kit (AddGene) and RVD repeat arrays were cloned into pCS2TAL3DD and pCS2TAL3RR (AddGene). Plasmids were transformed into DH5α competent cells (Invitrogen) and isolated using the QIAprep Spin Miniprep Kit (Qiagen).

Subsequently, plasmids encoding the scaffold and RVD arrays were linearized via restriction digest and 5′-capped mRNA was generated by *in vitro* transcription using the mMESSAGE mMACHINE SP6 Transcription Kit (Life Technologies). Capped mRNA was purified using the RNeasy Mini Kit (Qiagen). WT embryos were collected, and 40–50 pg of pooled left and right TALENs at equal concentrations were injected into the yolk of 1–4 cell stage embryos. Embryos were raised to adulthood, and sperm and progeny of these fish were collected and sequenced with MiSeq to screen for germline transmission.

### Western blot of Smyhc1 protein

For each sample, 20 embryos were collected and dechorionated in a 37°C 1 mg/ml pronase egg water solution. Embryos were anesthetized and deyolked by agitating in deyolking buffer (55 mM NaCl, 1.25 mM NaHCO₃, 1.8 mM KCl, 2.7 mM CaCl₂). Embryos were then washed twice in wash buffer (10 mM Tris pH 8.5,

110 mM NaCl, 3.5 mM KCl, 2.7 mM CaCl₂). Embryos were then agitated in RIPA with complete protease inhibitor cocktail (Roche Applied Sciences). 50 μg of each sample was loaded onto a sodium dodecyl sulfate–polyacrylamide gel electrophoresis (SDS–PAGE). Proteins were transferred to PVDF transfer membrane (Immobilon) for immunoblotting. Membranes were blocked in 5% skim milk for 1 h and incubated with antibody in 1% skim milk overnight at 4°C. The embryonic myosin heavy chain antibody [F59 (Iowa Developmental Studies Hybridoma bank)] (1:70) (Elworthy *et al*, 2008) was used to stain Smyhc1. A GAPDH antibody (Thermo Scientific) (1:500) was used as a loading control. Membranes were washed in 0.1% PBST (0.1% Tween-20 in PBS) and incubated with a secondary anti-mouse antibody conjugated to horseradish peroxidase and developed with a chemiluminescent substrate enhanced chemiluminescence (GE Healthcare).

### Genotyping of *smyhc1*$^{R673H}$ mutants

Individual fish were genotyped via DNA collection and PCR amplification of the *smyhc1* mutant region, with subsequent restriction digest targeting to identify the R673H allele. DNA was extracted from tissue samples in lysis buffer [0.1 M Tris, 0.005 M EDTA, 0.03% SDS, 0.2 M NaCl, 1% Proteinase K (New England Biolabs, P8107S)] and incubated at 50°C for 16 h and 95°C for 10 min. A fragment of DNA containing the R673H mutation was PCR-amplified. The amplified DNA was incubated with the restriction enzyme ApaLI in CutSmart buffer (New England BioLabs; R0507S, B7204S) for 16 h at 37°C, which cleaves *smyhc1* R673H mutant DNA fragments, allowing wild-type, homozygous, and heterozygous individuals to be identified after separation on an agarose gel.

### Alizarin red/Alcian blue skeletal staining

Adult zebrafish (1–2 years postfertilization, male and female) were euthanized and fixed in 4% paraformaldehyde in PBS at room temperature under agitation for 7 days. They were then incubated in acetone overnight under agitation at room temperature. After rinsing in water, they were then incubated overnight under agitation in staining solution (0.015% Alcian Blue, 0.005% Alizarin Red, 5% Glacial Acetic Acid, 59.5% Ethanol). The fish were rinsed once and then for 30 min under agitation at room temperature in water. The fish were incubated in 1% KOH under agitation at room temperature until the soft tissue became transparent, 1–3 weeks depending on the size of the fish. The 1% KOH solution was replaced periodically. The fish were stored in glycerol, and the scales removed manually if they had not fallen off during KOH clearing.

### Phalloidin-rhodamine/DAPI/Immunohistochemistry muscle staining and imaging

Zebrafish larvae 1–3 dpf were euthanized and fixed in 4% paraformaldehyde in PBS overnight under agitation at 4°C. The larvae were washed briefly and then twice for 30 min in PBD (0.1% Tween-20, 0.3% Triton X-100, 1% DMSO in PBS) under agitation at room temperature. The embryos were then incubated in block solution (0.1% Tween-20, 10% FBS, 2% BSA) at room temperature under agitation for 60 min. Primary antibodies were added to the block solution at the appropriate dilutions [F59 (Iowa

Developmental Studies Hybridoma bank) 1:10; anti-Caspase 3 (Sigma-Aldrich AB3623) 1:100; anti-α-actinin (Sigma-Aldrich A7811) 1:800; anti-phospho-SMAD2 (Sigma-Aldrich AB3849) 1:100] and incubated overnight under agitation for 4°C. The solution was removed, and the embryos were washed once quickly in PBD and twice for 30 min under agitation at room temperature. The embryos were then incubated in block solution for 60 min under agitation at room temperature. The appropriate secondary antibody was added to the block solution [Goat Anti-Rabbit IgG H&L (Alexa Fluor® 488) (Abcam ab150077) 1:500, Goat Anti-Mouse IgG (H&L) (Adsorbed Against: Hu., Bv., Hs.) (DyLight® 488) (Leinco M1331) 1:500], and the embryos were incubated at 4°C overnight under agitation in the dark. The solution was removed, and the embryos were washed once quickly in PBD and incubated with DAPI (Sigma-Aldrich D9542) at a dilution of 1:15,000 in PBST for 30 min under agitation at room temperature. The solution was removed, and the embryos were incubated in rhodamine phalloidin (Invitrogen 2009720) at a dilution of 1:1,000 in PBD for 30 min at room temperature. The embryos were then washed in PDB for 30 min at room temperature and mounted on slides in glycerol under cover slips before imaging under confocal microscopy. Confocal images of stained muscle were analyzed in ImageJ to measure myosepta distance.

## Microscopy

Confocal images were taken on an Olympus BX61WI fixed stage microscope with Olympus Fluoview FV1000 confocal laser scanning. Images were processed in ImageJ.

## Drug treatment

Embryos were randomly chosen for control or experimental treatment and dechorionated prior to drug treatment. Embryos were immersed in 40 μg/ml tricaine methanesulfonate in egg water (5 mM NaCl, 0.17 mM KCl, 0.33 mm CaCl$_2$, MgSO$_4$ in H$_2$O), 25 μM blebbistatin, or 25 μM para-aminoblebbistatin. Para-aminoblebbistatin in DMSO was mixed with egg water to facilitate dissolution of blebbistatin in aqueous buffer, generating 0.25% DMSO, 25 μM blebbistatin in egg water. For the tricaine treatment, the egg water was replaced every 24 h for 5 days. All incubations were performed in glass-bottom dishes. Images of the embryos were taken periodically.

## Noldus movement quantification

The Noldus DanioVision and EthoVision software were used to record and quantify movement of zebrafish larvae at 6 dpf. Larvae were acclimated in the DanioVision box for 5–10 min before using swimming data for analysis. Embryos were loaded into the DanioVision in 12-well cell culture plates in replicates and placed in wells at random. Five minutes of recorded behavioral data was analyzed for the distance traveled. The EthoVision software tracked and recorded fish movement data.

## Light stimulated motor activity

Light stimuli were administered to zebrafish embryos which were individually embedded in their chorions in an agarose mold. They were kept in an opaque box in a 28.5°C incubator between

### The paper explained

#### Problem

The debilitating joint contractures that are characteristic of distal arthrogryposis (DA) syndromes are currently suboptimally treated with supportive care. Understanding how dominant or recessive *MYH3* mutations cause contractures or bony fusions has been limited by poor access to human tissue, particularly during early development when *MYH3* is most highly expressed. Previous single-cell and small molecule studies suggest that DA mutations cause muscle hypercontraction, but vertebrate models are required to study the complex interactions between bone and muscle and to develop novel targeted therapeutics.

#### Results

Zebrafish carrying a single copy of the most common DA-associated *MYH3* substitution (R672H) displayed notochord bends that developed into scoliosis and vertebral fusions in adulthood, shortened sarcomeres and muscle fibers, and impaired swimming capacity. The direct chemical inhibition of muscle contraction with the myosin ATPase inhibitor para-aminoblebbistatin prevented the notochord bends from developing in both heterozygous and homozygous fish, suggesting that the mutant allele causes notochord and vertebral abnormalities through a mechanical increase in muscle tension.

#### Impact

We developed a viable zebrafish model of DA that is dually useful for both mechanistic studies and therapeutic drug development. Our work suggests that muscle hypercontractility mediated by the MYH3 mutation secondarily leads to vertebral fusions highlights the interconnectedness of the muscular and skeletal systems during early development. Furthermore, we have shown the beneficial effects of myosin ATPase inhibitors for the treatment of DA.

timepoints. At 17, 21, 24, 26, 28, 30, 32, 34, and 48 hpf, each mold was placed on a dissecting microscope stage, and a video recording was captured. The embryos were subjected to two cycles of 30 s in the dark and 30 s under light stimulus, with a final 30-s dark period afterward. The number of times each embryo twitched was counted in each recording.

## Swim tunnel quantification

Zebrafish were loaded into a swim tunnel with a steadily increasing flow. Once an individual fish fatigued, it lay against the back grate of the tunnel and was removed from the experiment to a separate tank, and the time to fatigue was recorded. After acclimating with no flow, the fish were subjected to 10 cm/s flow for 1 min. The flow then increased by 2 cm/s every minute until every fish fatigued.

## Statistical analysis

Differences in number of twitches, distance traveled, α-actinin intervals, and myoseptal intervals were quantified using nonparametric tests (Wilcoxon rank-sum test), so normal distribution was not assumed in the statistical tests. An estimate of variation was not included in the statistical analysis, but seems to vary between compared groups. R was used to analyze data and generate graphs.

Sample size was determined based on experience from previous findings and animal availability. Depending on the nature and

complexity of the assay, the maximum number of animals that could be tested was used to increase statistical power as much as possible. No samples were excluded from analysis. Zebrafish were chosen at random as subjects before genotype was determined. When determining survivability of zebrafish genotypes, entire clutches were genotyped to avoid selective bias. The investigator was not blinded before assessing skeletal, muscular, or gross anatomical abnormalities of zebrafish.

## Data availability

The zebrafish lines discussed in this paper are available upon request. No novel software was developed for the purposes of this study. No protein, DNA, or RNA sequence; Macromolecular structure; Crystallographic; Functional Genomics; or Proteomics and molecular interaction datasets were collected for the purposes of this study. This study includes no data deposited in external repositories.

**Expanded View** for this article is available online.

## Acknowledgements
Research reported in this publication was supported by National Institute of Arthritis and Musculoskeletal and Skin Diseases under Award Numbers R01AR067715 and R01AR070299, Eunice Kennedy Shriver National Institutes of Child Health and Human Development of the National Institutes of Health under the Award Number P01 HD084387, Washington University Institute of Clinical and Translational Sciences grant UL1 TR002345 from the National Center for Advancing Translational Sciences of the National Institutes of Health, Washington University Musculoskeletal Research Center (NIH/NIAMS P30 AR057235) (NIH/NIAMS P30 AR074992), and the Eunice Kennedy Shriver National Institute of Child Health & Human Development of the National Institutes of Health under Award Number P50 HD103525 to the Intellectual and Developmental Disabilities Research Center at Washington University. This study was funded with support from the University of Missouri Spinal Cord Injury Research Program, Shriners Hospital for Children, and the Children's Discovery Institute of Washington University and St Louis Children's Hospital.

## Author contributions
JW and LA wrote the first draft of the manuscript. JW, LA, and CAG designed the laboratory experiments, and analyzed and interpreted the data. JW, LA, MH, ZU, DSS, ANJ, MM, LS-K, MBD, and CAG participated in the analyses and interpretation of data, wrote or critically reviewed the manuscript, and reviewed and approved the final version.

## Conflict of interest
The authors declare that they have no conflict of interest.

## For more information
https://neuro.wustl.edu/labs/gurnett-dobbs-lab/

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
