## [Review Process File · EMBO Molecular Medicine]

MYH3-associated distal arthrogryposis zebrafish model is normalized with para-aminoblebbistatin

Julia Whittle, Lilian Antunes, Mya Harris, Zachary Upshaw, Diane Sepich, Aaron Johnson, Mayssa Mokalled, Lilianna Solnica-Krezel, Matthew Dobbs, and Christina Gurnett

DOI: [10.15252/emmm.202012356](https://doi.org/10.15252/emmm.202012356)

Corresponding author: Christina Gurnett (gurnettc@neuro.wustl.edu)

Review Timeline:

Submission Date:	19th Mar 20
Editorial Decision:	20th Apr 20
Revision Received:	20th Jul 20
Editorial Decision:	26th Aug 20
Revision Received:	8th Sep 20
Accepted:	9th Sep 20

Editor: Zeljko Durdevic

Transaction Report:

20th Apr 2020

Dear Prof. Gurnett,

Thank you for the submission of your manuscript to EMBO Molecular Medicine. We have now heard back from the three referees who agreed to evaluate your manuscript. As you will see from the reports below, the referees acknowledge the interest of the study. However, they raise some concerns that should be addressed in a major revision of the present manuscript. Addressing the reviewers' concerns in full will be necessary for further considering the manuscript in our journal.

Acceptance of the manuscript will entail a second round of review. Please note that EMBO Molecular Medicine encourages a single round of revision only and therefore, acceptance or rejection of the manuscript will depend on the completeness of your responses included in the next, final version of the manuscript. For this reason, and to save you from any frustrations in the end, I would strongly advise against returning an incomplete revision.

We realize that the current situation is exceptional on the account of the COVID-19/SARS-CoV-2 pandemic. Therefore, please let us know if you need more than three months to revise the manuscript.

I look forward to receiving your revised manuscript.

***** Reviewer's comments *****

Referee #1 (Remarks for Author):

In the manuscript by Whittle et al. the authors use zebrafish to model a complex skeletal syndrome known as distal arthrogyriposis (DA) which leads to joint contractures and scoliosis. DA is associated with mutations in the slow myosin heavy chain gene MYH3. However, how those mutations lead to the observed pathology is not well understood. The authors identified the zebrafish orthologue gene (*smyhc1*) and engineered both the equivalent of a common dominant mutation found in DA and a loss-of-function allele. Their analyses of these mutations implicate *smyhc1* in the assembly of slow muscle fibers in the axial skeleton that lead to kinking of the notochord, resulting in vertebral fusions and severe spine defects. Interestingly, the authors were able to partially rescue the embryonic defects of the dominant mutation via transient inhibition of myosin activity.

Overall, this is an interesting manuscript that represents a good example of disease modeling in zebrafish as it investigates the effect of a pathogenic mutation in processes and structures that are relevant for the disease. It is also important to note that authors made proper use of currently available technologies to generate a genetically sound model. The result shown in this manuscript provide mechanistic insight into the origin of spine defects in DA and offer a potential therapeutic avenue for this type of disorders.

The manuscript is generally well written and requires only a few corrections. However, it would benefit from further elaboration and more precision in some areas (see below). If available, it would be useful to add data documenting some specific findings or assumptions.

Major points:

1-The authors report spine defects for the *smyhc1*^{-/-} allele that develop between one two years of age. 1A-How common are those defects compared to that of WT or het siblings? 1B-given that defects occur in the caudal area and manifest in relatively old fish, a common occurrence in many genetic backgrounds, can those defects be tied to an embryonic defect that has no discernible phenotype early on? More data or a more cautious interpretation is needed.

Minor points:

2-Could the authors present RT data for the null allele?

3-Is there western blot data for the dominant mutation? hets would be most informative

4-Could the authors present or cite in situ hybridization data for *smyhc1*?

5-It would be useful to add a bit more information on the domain structure of MYH3/Smyhc1 and the basis of the identification of the orthologous mutation. Perhaps a multi-species alignment of the relevant domain could help.

Text issues:

6-In the intro the authors refer to the most common variant causing DA2A and DA2B is a R672C mutation, but then this is changed to R672H in the following paragraph, is this a typo?

7-The last sentence of the third section in Results, discussing the notochord "Despite the often severe..." is really difficult to read, please consider re-writing, perhaps splitting the sentence in two.

8-There is a typo "homozygoteosus" in the discussion.

Experimental suggestion:

While not strictly necessary, there is one experiment worth performing should it be technically possible.

The authors use transient treatment with para-aminobenzocysteamine to inhibit myosin activity, showing partial rescue of some of the most severe embryonic defects of the dominant allele. However, it is clear this treatment also has toxic side effects. Would it be possible to inject a cross of WT to *smyhc1*^{R673H/+} with a CRISPR pool targeting *smyhc1*? The expectation is that a number of fish

will effectively have a loss-of-function phenotype and mutations in cis to the dominant allele would rescue the most severe phenotypes, perhaps allowing the authors to examine whether spine defects can also be rescued.

Referee #2 (Remarks for Author):

The establishment of a zebra fish model for study of myosin mutations in skeletal muscle associated with congenital diseases is an important step, as vertebrate models are lacking. This is an important first step, however there are concerns that should be addressed to increase the potential impact of the study.

1. Blebbistatin compounds are known to be toxic, as pointed out by the authors. Indeed, in early stage muscle, these compounds have been shown to disrupt established sarcomeres and prevent sarcomere genesis. As such, it is unclear why these compounds were chosen. The Myokardia compound MYK-461 would appear to be a better choice, as it is less toxic and it, or its analogs, are being developed for treatment of cardiac muscle disease in humans. There are also other myosin inhibitors that should be considered.

2. Because of the problem with the myosin inhibitors used, clear establishment of muscle contraction with MYH3 R672H as contributing to, or being causal for, malformation, dysfunction and lethality has not been established. More rigorous effort is required.

3. It is misleading to call MYH3 R673H a 'gain of function' mutation. This mutation has been demonstrated (Racca 2015) to impair relaxation in human muscle samples, and specific force of individual muscle fibers was reduced.

4. If the thought is that the MYH3 R673H mutation is hyper-contractile, an assessment of muscle function should be done. This would strengthen the argument. In addition, a quantitative analysis of sarcomere lengths should be performed. This should be easily accomplished, especially in conditions as those presented in Figure 6. Along with this, higher magnification analysis, perhaps with electron microscopy, could reveal features of altered sarcomere formation that would increase the ability to make mechanistic interpretations.

5. A more quantitative analysis of the features in Figure 5 may also be helpful.

Referee #3 (Remarks for Author):

This paper adopts a zebrafish model to examine the phenotypic consequences of an allelic series of mutations at the *smyhc1* locus which is homologous to MYH3 in humans. A variety of phenotypes have been demonstrated in humans (caused by both biallelic and monoallelic genotypes) but the mechanistic relationship between them all remains obscure. This paper makes a useful contribution to this understanding in particularly noting the effects of adding null alleles in trans to a knock-in mutation R673H which has been recurrently shown to lead to a phenotype in humans characterised by a distal arthrogyrosis and scoliosis. The authors characterise their phenotypes using conventional morphometry in addition to light microscopy. Finally they show that addition of a drug p-aminoblebbistatin prevents the development of one of the more severe phenotypes expressed in their experimental system.

The paper is well written, explains the concepts and rationale for the experiments clearly and presents the relevant data in a logical and straightforward fashion. This work certainly adds to the understanding on this perplexing range of conditions and their mechanistic basis in humans. I have two major suggestions to improve the manuscript.

1. Although the work sets out to examine the consequences in muscle and bone of these various genotypes, the exploration of the muscle pathology is stronger than that for bone, which is primarily confined to the gross morphological preps to demonstrate vertebral fusions. In mice, Zeiba et al have shown some evidence for a mechanistic basis for vertebral fusions in *Myh3*^{-/-} mice that relates to re-direction of differentiation programmes in the intervertebral disks. Although some of the genotypes did (and some did not apparently, although I am not so convinced that authors can be categorical about this in what is depicted in Fig 1B; *smyhc1*^{-/-}) show intervertebral fusions, linking these observations with those in the paper by Zeiba is not attempted. Some authors contend that *Myh3* is expressed in bone and that the notochord phenotype is a cell intrinsic result of this, in contrast to it being a biomechanical result of myofiber induced forces on these structures. It would be very useful to have the authors address this issue.

It would be illuminating to understand if there is a mechanistic continuum across the genotypes examined here that is sponsored by the aberrations in TGF-beta (as shown by Zeiba et al) noted by them in this discussion. These data are particularly important in linking the work described here to the observation of a fusion phenotype in humans in DA8/SCTS.

2. The authors present some suggestive histological findings in muscle from their fish that suggests that the *smyhc1*R673H alleles lead to some disintegrity at the sarcomeric level (Fig 3A/B). This is intriguing because the genetics of the analogous conditions in humans does point to this allele being some kind of gain of function and the genetic data in this paper reinforce this contention. It would strengthen this paper if the anomalous structure of the sarcomere in these fish is teased out more. Some immunohistochemistry examining Z-disc structure for instance would be helpful as would stains to examine the hypothesis that the histological appearances might primarily result in cell death and degeneration of sarcomeric function in general. Presently, the current data stops at a frustratingly premature point!

Minor suggestions

1. Some of the linking conclusions presented in the results sections could be expressed more cautiously. For instance, the statement "We conclude that larval notochord abnormalities predispose *smyhc1*R673H/+ fish to vertebral fusions in adulthood" (pg 9) implies a causative link between two observations noted in this morphological series of observations where in fact this is only one possible conclusion from noting this association. Similarly, the last statement in the results section about the mode of action of p-aminoblebbistatin leaps straight to a mechanistic conclusion. Of course, this is one (very exciting) potential conclusion but a firm conclusion that this is the case is premature in the absence of experiments that would demonstrate that the effect of the drug is specific, obeys a dose response etc etc. I am not suggesting that these experiments need to be performed but more that the conclusions couched more cautiously.

p.14 homozygotesous fish

p15. The summary of the association of the human alleles that their associated phenotypes is not quite right, in part due to some shortcomings of the published literature. In all cases of SCTS caused by fully characterised bi-allelic genotypes, those genotypes have consisted of a haploinsufficient allele in trans with a hypomorphic loss of function allele. Dominantly inherited SCTS (and some sporadic cases) are always caused by missense alleles (likely conferring some kind of dominant negative effect). Some cases that appear to breach these rules only appear so because

they were incompletely characterised or the hypomorphic allele described by Cameron-Christie et al was not sought.

Reviewer's Comments to Author:

Referee #1

Comments to the Author:

In the manuscript by Whittle et al. the authors use zebrafish to model a complex skeletal syndrome known as distal arthrogryposis (DA) which leads to joint contractures and scoliosis. DA is associated with mutations in the slow myosin heavy chain gene MYH3. However, how those mutations lead to the observed pathology is not well understood. The authors identified the zebrafish orthologue gene (*smyhc1*) and engineered both the equivalent of a common dominant mutation found in DA and a loss-of-function allele. Their analyses of these mutations implicate *smyhc1* in the assembly of slow muscle fibers in the axial skeleton that lead to kinking of the notochord, resulting in vertebral fusions and severe spine defects. Interestingly, the authors were able to partially rescue the embryonic defects of the dominant mutation via transient inhibition of myosin activity.

Overall, this is an interesting manuscript that represents a good example of disease modeling in zebrafish as it investigates the effect of a pathogenic mutation in processes and structures that are relevant for the disease. It is also important to note that authors made proper use of currently available technologies to generate a genetically sound model. The result shown in this manuscript provide mechanistic insight into the origin of spine defects in DA and offer a potential therapeutic avenue for this type of disorders.

The manuscript is generally well written and requires only a few corrections. However, it would benefit from further elaboration and more precision in some areas (see below). If available, it would be useful to add data documenting some specific findings or assumptions.

Major points:

1-The authors report spine defects for the *smyhc1*^{-/-} allele that develop between one two years of age.
1A-How common are those defects compared to that of WT or het siblings?

Quantification of spinal defect prevalence have been added to the manuscript in Figure 4.

1B-given that defects occur in the caudal area and manifest in relatively old fish, a common occurrence in many genetic backgrounds, can those defects be tied to an embryonic defect that has no discernible phenotype early on? More data or a more cautious interpretation is needed.

We agree that this is a possibility- we also agree that this would be interesting to look at, but believe it is outside the scope of the paper. We have amended the text to a more explicitly cautious interpretation of the presented data.

Minor points:

2-Could the authors present RT data for the null allele?

We show that the *smyhc1* protein is not present in *smyhc1*^{-/-} fish by western Blot, and have added immunohistochemical data to Figure 1 showing its absence.

3-Is there western blot data for the dominant mutation? hets would be most informative

Due to the difficulty acquiring sufficient tissue from individually genotyped embryos for western blot for the dominant mutation, we have instead included immunohistochemistry data in Figure 1 that shows the presence of Smyhc1 protein staining within the muscles of smyhc1R673H/+ and smyhc1R673H/R673H fish.

4-Could the authors present or cite in situ hybridization data for smyhc1?

We added explicit reference to existing in situ hybridization data (Rauch et al., 2003).

5-It would be useful to add a bit more information on the domain structure of MYH3/Smyhc1 and the basis of the identification of the orthologous mutation. Perhaps a multi-species alignment of the relevant domain could help.

An alignment/domain schematic was added to Figure 1.

Text issues:

6-In the intro the authors refer to the most common variant causing DA2A and DA2B is a R672C mutation, but then this is changed to R672H in the following paragraph, is this a typo?

Thank you for catching this error, it has now been changed to reflect the prevalence of both variants.

7-The last sentence of the third section in Results, discussing the notochord "Despite the often severe..." is really difficult to read, please consider re-writing, perhaps splitting the sentence in two.

We have clarified the wording of this sentence.

8-There is a typo "hmozygoteosus" in the discussion.

Thank you for catching this. We have corrected the spelling.

Experimental suggestion:

While not strictly necessary, there is one experiment worth performing should it be technically possible.

The authors use transient treatment with para-aminobenzocysteamine to inhibit myosin activity, showing partial rescue of some of the most severe embryonic defects of the dominant allele. However, it is clear this treatment also has toxic side effects. Would it be possible to inject a cross of WT to smyhc1R673H/+ with a CRISPR pool targeting smyhc1? The expectation is that a number of fish will effectively have a loss-of-function phenotype and mutations in cis to the dominant allele would rescue the most severe phenotypes, perhaps allowing the authors to examine whether spine defects can also be rescued.

This data can be acquired, and we agree that this experiment would be useful. However, we believe this experiment remains outside the scope of the current paper; we believe the experiment is worth exploring in future research endeavors.

Referee #2

Remarks for Author:

The establishment of a zebra fish model for study of myosin mutations in skeletal muscle associated with congenital diseases is an important step, as vertebrate models are lacking. This is an important first step, however there are concerns that should be addressed to increase the potential impact of the study.

1. Blebbistatin compounds are known to be toxic, as pointed out by the authors. Indeed, in early stage muscle, these compounds have been shown to disrupt established sarcomeres and prevent sarcomere genesis. As such, it is unclear why these compounds were chosen. The Myokardia compound MYK-461 would appear to be a better choice, as it is less toxic and it, or its analogs, are being developed for treatment of cardiac muscle disease in humans. There are also other myosin inhibitors that should be considered.

Unfortunately, despite lengthy discussions with Myokardia, we were unable to receive any of this compound to evaluate in our model. Therefore, we proceeded with compounds which were readily accessible. Because blebbistatin compounds were previously demonstrated to be effective in zebrafish skeletal muscle by directly inhibiting actin-myosin interaction, we started with these. We agree that our model will be ideally suited to evaluate new molecules that have less toxicity as they are being developed.

2. Because of the problem with the myosin inhibitors used, clear establishment of muscle contraction with MYH3 R672H as contributing to, or being causal for, malformation, dysfunction and lethality has not been established. More rigorous effort is required.

We agree that a direct measurement of muscle tension would be very useful- however we believe this is outside the scope of our paper. To complement our manuscript, we have generated additional data, including sarcomere length, in our mutant fish to further characterize the effect of the variants on muscle function.

3. It is misleading to call MYH3 R673H a 'gain of function' mutation. This mutation has been demonstrated (Racca 2015) to impair relaxation in human muscle samples, and specific force of individual muscle fibers was reduced.

The wording of the mutation effect has been amended in the text to be more precise in its description in the third paragraph of the introduction.

4. If the thought is that the MYH3 R673H mutation is hyper-contractile, an assessment of muscle function should be done. This would strengthen the argument. In addition, a quantitative analysis of sarcomere lengths should be performed. This should be easily accomplished, especially in conditions as those presented in Figure 6. Along with this, higher magnification analysis, perhaps with electron microscopy, could reveal features of altered sarcomere formation that would increase the ability to make mechanistic interpretations.

We appreciate the suggestion, and have determined sarcomere lengths using higher magnification analysis and added this to Figure 6. We also performed additional DAPI and anti- α -actinin stains to further characterize the effect of the mutations on muscle in greater detail.

5. A more quantitative analysis of the features in Figure 5 may also be helpful.

Quantitative analysis of Figure 5A was conducted and data were added to Figure 5.

Referee #3

Remarks for Author:

This paper adopts a zebrafish model to examine the phenotypic consequences of an allelic series of mutations at the *smyhc1* locus which is homologous to MYH3 in humans. A variety of phenotypes have been demonstrated in humans (caused by both biallelic and monoallelic genotypes) but the mechanistic relationship between them all remains obscure. This paper makes a useful contribution to this understanding in particularly noting the effects of adding null alleles in trans to a knock-in mutation R673H which has been recurrently shown to lead to a phenotype in humans characterised by a distal arthrogyrosis and scoliosis. The authors characterise their phenotypes using conventional morphometry in addition to light microscopy. Finally they show that addition of a drug p-aminobisphosphonate prevents the development of one of the more severe phenotypes expressed in their experimental system.

The paper is well written, explains the concepts and rationale for the experiments clearly and presents the relevant data in a logical and straightforward fashion. This work certainly adds to the understanding on this perplexing range of conditions and their mechanistic basis in humans. I have two major suggestions to improve the manuscript.

1. Although the work sets out to examine the consequences in muscle and bone of these various genotypes, the exploration of the muscle pathology is stronger than that for bone, which is primarily confined to the gross morphological preps to demonstrate vertebral fusions. In mice, Zeiba et al have shown some evidence for a mechanistic basis for vertebral fusions in *Myh3*^{-/-} mice that relates to re-direction of differentiation programmes in the intervertebral disks. Although some of the genotypes did (and some did not apparently, although I am not so convinced that authors can be categorical about this in what is depicted in Fig 1B; *smyhc1*^{-/-}) show intervertebral fusions, linking these observations with those in the paper by Zeiba is not attempted. Some authors contend that *Myh3* is expressed in bone and that the notochord phenotype is a cell intrinsic result of this, in contrast to it being a biomechanical result of myofiber induced forces on these structures. It would be very useful to have the authors address this issue.

It would be illuminating to understand if there is a mechanistic continuum across the genotypes examined here that is sponsored by the aberrations in TGF-beta (as shown by Zeiba et al) noted by them in this discussion. These data are particularly important in linking the work described here to the observation of a fusion phenotype in humans in DA8/SCTS.

We share the reviewer's interest in the effects of these mutants on bone development. On closer review of the alizarin stained bone of our *smyhc1*^{R673H/+} fish, it does appear that bone fusions are present even in regions where the spine is straight, suggesting that the variant may lead to fusions in the absence of notochord kinking. The discussion has been revised to reflect this.

2. The authors present some suggestive histological findings in muscle from their fish that suggests that the *smyhc1*^{R673H} alleles lead to some disintegrity at the sarcomeric level (Fig 3A/B). This is intriguing because the genetics of the analogous conditions in humans does point to this allele being some kind of gain of function and the genetic data in this paper reinforce this contention. It would strengthen this paper if the anomalous structure of the sarcomere in these fish is teased out more. Some immunohistochemistry examining Z-disc structure for instance would be helpful as would stains to examine the hypothesis that the histological appearances might primarily result in cell death and degeneration of sarcomeric function in general. Presently, the current data stops at a frustratingly premature point!

We appreciate this suggestion. Therefore, we performed additional histology to evaluate the role of *smyhc1* on muscle development, and after staining for α -actinin, found that *smyhc1* is required for the proper development of the Z-disc (Figure 6C,D). Indeed, as the reviewer smartly predicted, cell death does appear to be involved in the histological appearance of the *smyhc1*^{R673H/+} and *smyhc1*^{R673H/+} muscle, as shown by Caspase-3 staining (Figure EV4).

Minor suggestions

1. Some of the linking conclusions presented in the results sections could be expressed more cautiously. For instance, the statement "We conclude that larval notochord abnormalities predispose *smyhc1*^{R673H/+} fish to vertebral fusions in adulthood" (pg 9) implies a causative link between two observations noted in this morphological series of observations where in fact this is only one possible conclusion from noting this association. Similarly, the last statement in the results section about the mode of action of p-aminobisphosphonate leaps straight to a mechanistic conclusion. Of course, this is one (very exciting) potential conclusion but a firm conclusion that this is the case is premature in the absence of experiments that would demonstrate that the effect of the drug is specific, obeys a dose response etc etc. I am not suggesting that these experiments need to be performed but more that the conclusions couched more cautiously.

The wording of these conclusions have been modified to yield a more cautious interpretation of the results.

p.14 homozygotesous fish

Thank you for catching this, the spelling has been corrected.

p15. The summary of the association of the human alleles that their associated phenotypes is not quite right, in part due to some shortcomings of the published literature. In all cases of SCTS caused by fully characterised bi-allelic genotypes, those genotypes have consisted of a haploinsufficient allele in trans with a hypomorphic loss of function allele. Dominantly inherited SCTS (and some sporadic cases) are always caused by missense alleles (likely conferring some kind of dominant negative effect). Some cases that appear to breach these rules only appear so because they were incompletely characterised or the hypomorphic allele described by Cameron-Christie et al was not sought.

Thank you for pointing this out, description of the literature has been amended to more reflective of previous research.

26th Aug 2020

Dear Prof. Gurnett,

Thank you for the submission of your revised manuscript to EMBO Molecular Medicine. We have now received the enclosed reports from the referees that were asked to re-assess it. As you will see the reviewers are now globally supportive and I am pleased to inform you that we will be able to accept your manuscript pending the following final amendments:

1) Please discuss the possible toxic effects of blebbistatin and how this may have influenced results of the study as suggested by the referee #2.

***** Reviewer's comments *****

Referee #2 (Comments on Novelty/Model System for Author):

The study would be improved by measurements of muscle contraction, this was suggested in the initial review. It is OK without, but impact is not as high as it could be.

Referee #2 (Remarks for Author):

The authors should discuss the possible toxic effects of blebbistatin and how this may have influenced results

Referee #3 (Comments on Novelty/Model System for Author):

The model addresses mechanism for these alleles that have not yet been addressed in a vertebrate. The muscle phenotypes are addressed well and the skeletal phenotypes are described but not mechanistically dissected

Referee #3 (Remarks for Author):

My concerns have been addressed adequately.

Editor's Response

1) Please discuss the possible toxic effects of blebbistatin and how this may have influenced results of the study as suggested by the referee #2.

This has been added to the manuscript in the results section.

Reviewer's comments

Referee #2 (Comments on Novelty/Model System for Author):

The study would be improved by measurements of muscle contraction, this was suggested in the initial review. It is OK without, but impact is not as high as it could be.

Thank you for the comment. We will work to study muscle contraction in future publications.

Referee #2 (Remarks for Author):

The authors should discuss the possible toxic effects of blebbistatin and how this may have influenced results

This has been added to the results section.

Referee #3 (Comments on Novelty/Model System for Author):

The model addresses mechanism for these alleles that have not yet been addressed in a vertebrate. The muscle phenotypes are addressed well and the skeletal phenotypes are described but not mechanistically dissected

Thank you for the comment. We are working on exploring the skeletal effects further in future publications.

Referee #3 (Remarks for Author):

My concerns have been addressed adequately.

The authors performed the requested changes.

Corresponding Author Name: Christina Gurnett

Manuscript Number: EMM-2020-12356-V3